# The Burden and Risk Factors of Patellar and Achilles Tendinopathy in Youth Basketball: A Cohort Study

**DOI:** 10.3390/ijerph18189480

**Published:** 2021-09-08

**Authors:** Oluwatoyosi B. A. Owoeye, Luz Palacios-Derflingher, Kati Pasanen, Tate HubkaRao, Preston Wiley, Carolyn A. Emery

**Affiliations:** 1Department of Physical Therapy and Athletic Training, Doisy College of Health Sciences, Saint Louis University, St. Louis, MO 63104, USA; 2Sport Injury Prevention Research Centre, Faculty of Kinesiology, University of Calgary, Calgary, AB T2N 1N4, Canada; lmpalaci@ucalgary.ca (L.P.-D.); kati.pasanen@ucalgary.ca (K.P.); wiley@ucalgary.ca (P.W.); caemery@ucalgary.ca (C.A.E.); 3Department of Community Health Sciences, Cumming School of Medicine, University of Calgary, Calgary, AB T2N 4Z6, Canada; tate.hubkarao@ucalgary.ca; 4Alberta Children’s Hospital Research Institute, University of Calgary, Calgary, AB T3B 6A8, Canada; 5Tampere Research Center of Sport Medicine, UKK Institute, FIN-33501 Tampere, Finland; 6McCaig Institute for Bone and Joint Health, University of Calgary, Calgary, AB T2N 4Z6, Canada; 7Department of Pediatrics, Cumming School of Medicine, University of Calgary, Calgary, AB T3B 6A8, Canada; 8Faculty of Kinesiology, Sport Medicine Centre, University of Calgary, Calgary, AB T2N 1N4, Canada; 9O’Brien Institute for Public Health, Calgary, AB T2N 4Z6, Canada

**Keywords:** jumpers’ knee, tendon pain, overuse injury, prevention, youth sport

## Abstract

This study aimed at evaluating the burden and risk factors of patellar and Achilles tendinopathy among youth basketball players. Patellar and Achilles tendinopathy were prospectively monitored in 515 eligible male and female youth basketball players (11–18 years) through a competitive season. Overall, the season prevalence of patellar tendinopathy was 19.0% (95% CI: 15.7–22.7%), 23.2% (95% CI: 18.6–28.2%) in males and 12.5% (95% CI: 8.3–17.9%) in females. The season prevalence of Achilles tendinopathy was 4.3% (95% CI: 2.7–6.4%), 4.1% (95% CI: 2.2–7.0%) in males and 4.5% (95% CI: 2.1–8.4%) in females. Median proportion of symptoms duration was 83% of average total weeks of basketball exposure for patellar tendinopathy and 75% for Achilles tendinopathy. Median time to patellar tendinopathy onset was 8 weeks for male players and 6 weeks for female players. Higher odds of patellar tendinopathy risk were seen in males (OR: 2.23, 95% CI: 1.10–4.69) and players with previous anterior knee pain had significantly elevated odds (OR: 8.5, 95% CI: 4.58–16.89). The burden and risk of patellar tendinopathy is high among competitive youth basketball players. Risk factors include sex and previous anterior knee pain. These findings provide directions for practice and future research.

## 1. Introduction

Patellar tendinopathy (PTP) is a common overuse/gradual onset injury in elite-level basketball with a reported prevalence of 32% (60% career prevalence) [1]. Although the prevalence of Achilles tendinopathy (ATP) is relatively low, its impact on elite adult basketball players is concerning [2]. Tendinopathy is primarily a clinical problem of pain with or without dysfunction [3]. Individuals may also have tendon pathology, which is a primary factor for tendon degeneration, disrepair, and potential rupture [4,5], and this may result in early retirement from professional sport and potentially impact players’ long-term health [2,5].

Despite the potentially significant impact of PTP and ATP on competitive athletes, there is a paucity of literature regarding the burden and risk factors of PTP and ATP, particularly in youth sport [6,7]. Only one study [8], conducted several years ago, investigated the prevalence of PTP in youth basketball players, and there is currently no single study on the prevalence of ATP in youth basketball. A prevalence of 7% (11% in males and 2% in females) was previously reported in a cross-sectional study through clinical evaluation, limiting the potential capture of all PTP in youth basketball players [8]. 

To fully understand the risk of injury, a combination of measures of risk has been advocated [9]. Systematic reviews on risk factors for PTP in athletes (mostly > 18 years) concluded that there is currently a lack of strong evidence evaluating risk factors for PTP and have suggested further research in prospective studies [10]. No single study has investigated the burden (e.g., symptoms duration) of and time to PTP and ATP in youth basketball. In order to have a robust understanding of risk, it is essential that conventional measures of risk are supplemented with injury burden in epidemiological studies [9].

Tendinopathies are often chronic in nature, with periods of remission and exacerbation. Therefore, prevalence (the proportion of athletes affected at a given time), and not incidence rate (new cases over a given time), is the appropriate measure of risk [11]. Conventional injury surveillance tools, ideally suited to capture acute injuries, are less appropriate and insensitive to capture the magnitude and severity of PTP and ATP [11,12]. This clinical and research challenge may be overcome using a novel, validated injury surveillance method—the Oslo Sports Trauma Research Centre (OSTRC) Overuse Injury Questionnaire—to accurately register overuse injuries in sport using athlete self-reporting [12]. The OSTRC Questionnaire is, however, limited in its scope of application for specific injury types and diagnosis as it does not provide information about individual overuse injuries. To address this limitation, the OSTRC Overuse Injury Questionnaire was adapted by our team and validated against clinical evaluation for self-reporting PTP [13]. 

The approach to clinical management of PTP and ATP is variable and treatment modalities have limited success in patients with chronic tendinopathy [14,15,16]. Prevention and early diagnosis are key to protecting players’ health and avoiding long-term consequences. Thus, the most expedient population to implement the prevention of PTP and ATP and associated consequences is youth athletes. The objectives of this study were: (1) to evaluate the burden of PTP and the burden of ATP, including prevalence, time to tendinopathy (first report), and symptoms duration, and (2) to evaluate risk factors associated with PTP and ATP in competitive youth male and female basketball players.

## 2. Materials and Methods

### 2.1. Study Design and Participants

The STROBE guidelines for reporting cohort studies were followed [17]. We conducted a prospective cohort study involving adolescent players from high school (December 2016–March 2017) and club basketball (March 2017–June 2017) teams in Calgary, Canada. A total of 52 high schools and 23 basketball clubs in Calgary, Canada, were invited to participate in this study. Prior to recruiting participants from high school basketball teams, consent was obtained from school principals followed by the Physical Education directors and basketball coaches. Similarly, prior to recruiting participants from clubs, consent was obtained from club managers followed by coaches. 

Players were eligible to participate if they were formally registered with their school or club basketball team. Eligibility to participate in the study was irrespective of anterior knee pain, PTP or ATP at baseline; it was expected that a few players would have ongoing PTP and ATP at time of enrollment and excluding players reporting PTP and ATP at baseline would result in a biased study population [11,18]. This methodological approach underpins the use of prevalence measures rather than incidence [11]. Exclusion criteria at baseline included acute lower extremity injuries or medical conditions (including those that may be associated with tendon problems, e.g., Type II diabetes) precluding competitive basketball participation or baseline performance testing. Written informed consent was requested from all players (in cases of players younger than 15 years, written parental consent and child assent were obtained). This study was approved by the Conjoint Health Research Ethics Board of the University of Calgary, Alberta, Canada (REB16-0864). 

### 2.2. Registration of Patellar and Achilles Tendinopathy

Main study outcomes were season prevalence of PTP and season prevalence of ATP. Diagnosis of PTP and ATP was based on self-report by participants using the Oslo Sports Trauma Research Centre Patellar [13] and Achilles Tendinopathy Questionnaire, an adaptation of the OSTRC Overuse Injury Questionnaire [19]. This is a two-part questionnaire with similar but specific questions relating to the knee and ankle joints (Appendix A). The patellar tendinopathy part of the questionnaire, the OSTRC-Patellar Tendinopathy Questionnaire, has been validated against clinical evaluation in youth basketball players with an overall sensitivity of 79% and specificity of 98% [13].

Players were asked to complete the OSTRC Tendinopathy Questionnaire weekly online using a mobile smartphone or alternatively using an identical hard-copy paper version handed to them by their team designate (research assistant, team student trainer, or coach). Players with smartphones were prompted to complete the OSTRC Tendinopathy Questionnaire weekly (every Sunday evening) and a reminder was automatically sent after the following day if the questionnaire was not completed. For players who opted to complete the paper version of the questionnaire, it was distributed every Monday at the time of practice. 

Determined based on teams’ final league schedules, as available online, for the 2016/2017 league seasons, players with less than the expected number of OSTRC Tendinopathy Questionnaire responses through the competitive season (i.e., 10–12 weeks of high school basketball season and 8–14 weeks of club season, depending on whether the player/team made playoffs) were followed up post-season through a telephone interview by the study physical therapist or trained research assistants to complete the OSTRC Tendinopathy Questionnaire for any missing weeks. The follow-up OSTRC Tendinopathy Questionnaire contained the same questions as the weekly questionnaire, except for the questions relating to weekly severity scores. Players reporting PTP or ATP were also asked to estimate the number of weeks they experienced symptoms (pain with/without dysfunction) during the season in the follow-up phone interview. 

### 2.3. Measures of Burden

#### 2.3.1. Prevalence Measures

We calculated season prevalence of PTP by dividing the number of players reporting at least one episode of PTP through the season by the number of players in the study. This calculation was performed for all players and separately for male and female players. Analogous calculations were performed for prevalence of ATP. Weekly prevalence was also calculated for all and substantial tendinopathy (for each week, number of players reporting, and episode over the number of players answering the questionnaire). Substantial tendinopathy connotes PTP or ATP leading to moderate or severe reductions in training volume or performance, or an inability to participate (that is, responses c, d, or e in either question #2 or #3 on the OSTRC Tendinopathy Questionnaire) (see Appendix A) [12].

#### 2.3.2. Time to Tendinopathy Onset

We reported time to tendinopathy onset during the study, in weeks, for every first report of PTP or ATP. This was based on a specific question on the OSTRC Tendinopathy Questionnaire—question #6 for each part—asking if a player’s reported symptoms were experienced for the first time the previous week (see Appendix A).

#### 2.3.3. Symptoms Duration and Severity 

Measures of PTP and ATP burden on players reporting tendinopathy included proportion of symptoms duration (percentage of weeks with symptoms) and level of dysfunction based on players’ weekly severity scores. Severity scores (0–100 in arbitrary units [AU]) were derived from the fourth set of questions on the OSTRC Tendinopathy Questionnaire and calculated as advised by Clarsen et al. [12] in the original OSTRC Overuse Injury Questionnaire. Time to tendinopathy and severity scores were only obtained in players who completed the weekly OSTRC Tendinopathy Questionnaire.

### 2.4. Potential Risk Factors

All participants were required to complete a pre-participation questionnaire at the time of enrollment in study. Potential risk factors considered included demographic and sport-related factors such as age, sex, body weight, height, league setting (school vs. club basketball league), previous knee injury (1-year history), previous anterior knee pain (3-month history), playing position, and basketball specialization (single-sport basketball participation based on baseline information of whether a player was involved in only basketball or multiple sports in the previous year). Our choice of potential risk factors was informed by previous literature [10,20]. 

### 2.5. Statistical Analysis

Statistical analyses were performed using Stata (StataCorp LP, College Station, TX, USA, version 14.4) and Microsoft Excel for Mac (version 16.16.1). Player characteristics were described with means and standard deviations (SD) for numerical variables, or median (range) (if not normally distributed), or frequencies and proportions (%) for categorical variables, by sex. For players participating in both school and club seasons, an assumption of independence was made, given that a small proportion of players participated in both seasons. 

Season prevalence estimates were reported with exact 95% confidence intervals (CI) for all players and by sex. Weekly prevalence (all and substantial tendinopathy) were plotted over time to identify trends throughout the course of the study. Time to tendinopathy (i.e., week of first tendinopathy report) was summarized using median and interquartile range (IQR). Symptoms duration was estimated in proportion (%), that is, the number of weeks with symptoms divided by the number of participation weeks for each player, and the median (first (Q1) and third (Q3) quartiles) of these percentages was calculated. The weekly severity scores reported by players (ones that completed one or more weekly OSTRC Tendinopathy Questionnaire) with tendinopathy were summarized using median for each week and plotted over time to identify trends. 

Some potential risk factor variables had substantial missing data, so we implemented the multivariate imputation by chained equations (MICE) to impute missing data. This algorithm augments for perfect prediction during imputation of data and adjusts for differences in variable types (i.e., binary, nominal and continuous variables) by modelling each variable according to its distribution [21,22]. Considering the extent of missing data, we generated 30 iterations of imputed datasets for optimal regression modelling [22]. 

Based on season prevalence of PTP and complete MICE datasets, a multivariable logistic regression analysis adjusted for team clusters was used to estimate odds ratios (ORs) and 95% CI to evaluate risk factors for PTP. The following covariates were included from the beginning: height, previous knee injury, playing position, and basketball specialization. A backward elimination regression analysis regression technique was employed. Covariates with no significant effects were removed from the model, one at a time, considering statistical (0.05 significance level) and clinical significance. Covariates were also checked for a change in more than 10% alteration in other coefficients. 

Although we had an a priori plan to conduct a multivariable logistic regression analysis for ATP risk factor evaluation, the total number of participants with ATP outcomes were too few to generate any meaningful model. 

## 3. Results

### 3.1. Player Characteristics

A total of 515 players from 63 teams completed this study: 315 (61.4%) males and 200 (38.6%) females. The selection and flow of players through the study are presented in Figure 1. The distribution of player baseline characteristics by sex is presented in Table 1. Thirteen players (2.5%; 8 males, 5 females) in this cohort participated in both school and club seasons—an assumption of independence was made in these players given the small proportion. 

### 3.2. Response Rate

All 515 players completed at least one version of the OSTRC Tendinopathy Questionnaire (either in-season or at follow-up), yielding an overall response rate of 100%. However, 350 (68%) of all players completed one or more (in-season) weekly OSTRC Tendinopathy Questionnaire. The average weekly response rate to the weekly OSTRC Tendinopathy Questionnaire was 62%. Characteristics (e.g., sex, age, and league setting) of the players that responded (having one or more weekly questionnaire response) and non-responders to the weekly questionnaire were similar. 

### 3.3. Prevalence of Patellar and Achilles Tendinopathy

A total of 98 players reported at least one episode of PTP through the league season (spanning an average of 9.5 weeks), equivalent to an overall season prevalence of 19.0% (95% CI: 15.7–22.7%), 23.2% (95% CI: 18.6–28.2%) in males (*n* = 73) and 12.5% (95% CI: 8.3–17.9%) in females (*n* = 25). Forty-five out of 98 players (48%) reporting PTP reported bilateral PTP, 29 males (39%) and 16 females (64%). For ATP, 22 players reported at least one episode of ATP through the league season, resulting in a season prevalence of 4.3% (95% CI: 2.7–6.4%), 4.1% (95% CI: 2.2–7.0%) in males (*n* = 13) and 4.5% (95% CI: 2.1–8.4%) in females (*n* = 9). Ten (45%) of the 22 players reporting ATP had bilateral ATP: five males (38%) and five females (56%).

Trends in the weekly prevalence of PTP and ATP are presented in Figure 2. While the prevalence of all PTP increased slightly over the course of the season in all players, the prevalence of all ATP was within the same range. The prevalence of substantial PTP and ATP ranged between 0% and 5% in both male and female players through the season.

### 3.4. Time to Tendinopathy Onset

The median (IQR) time to onset of PTP was 7 (4) weeks overall, 8 (4) weeks for males and 6 (5) weeks for females. No new cases of ATP were reported through the season; athletes either reported the same pain as the previous week or a return of pain that had gone away.

### 3.5. Symptoms’ Duration and Severity 

For players reporting tendinopathy (*n* = 98), the median (Q1, Q3) of the symptoms’ duration proportion for PTP was 83% (25%, 100%), 83% (27%, 100%) in males and 98% (25%, 100%) in females. The median proportion (Q1, Q3) of symptoms’ duration for ATP was 75% (25%, 100%), 100% (25%, 100%) in males and 63% (35%, 88%) in females. The overall severity score ranged from 17–68AU for PTP and 0–60AU for ATP.

### 3.6. Multivariable Analysis of Risk Factors

The final multivariable logistic regression model showed that the odds of PTP in males was 2.23 (95% CI: 1.10–4.69) times the odds of PTP in females; the odds of PTP in players with previous anterior knee pain was 8.79 (95% CI: 4.58–16.89) times the odds of players reporting no anterior knee pain (Table 2). 

## 4. Discussion

In this study, we evaluated the prevalence, time to tendinopathy, and symptoms’ duration of PTP and ATP in youth basketball players and examined associated risk factors for PTP in a cluster-adjusted multivariable regression analysis. To our knowledge, this is the first study to prospectively report the burden and risk factors of PTP in youth basketball. 

### 4.1. Prevalence of Patellar and Achilles Tendinopathy

Overall, we found a high prevalence of PTP and a relatively low prevalence of ATP. Our findings suggest that PTP is about five times as common as ATP among youth basketball players. While there was no difference in the season prevalence of ATP between male and female players, we found a significant difference in the season prevalence of PTP for male and female players. Our results showed that PTP became increasingly common among players as the season progressed, but the prevalence of ATP remained within the same range. Studies relating to the prevalence and risk factors of PTP and ATP in basketball are sparse and there is currently no study reporting the prevalence of ATP in youth basketball; the potential for comparison with other studies is therefore limited. An overall PTP prevalence of 19% (23% in males and 13% in females) presented in our study is lower than the prevalence (32%) reported for male elite adult basketball players [1]. However, our PTP prevalence estimates more than double the 7% (11% in males and 2% in females) previously reported by Cook et al. in youth basketball players aged 13–18 years [8]. Differences in study designs between our study and that of Cook et al. may explain, in part, the large difference in prevalence estimates. For example, in our study, we prospectively examined PTP weekly (repeated measurements) through self-reporting (which is highly sensitive in capturing gradual onset injuries) [23,24] to derive our main outcome of season prevalence of PTP, while in the study by Cook et al., PTP prevalence was based on a cross-sectional examination (one-time measurement) by a clinician. Further to this, the frequency and intensity of participation appears to have increased over the past decade, and it is very likely gradual onset/overuse injuries have increased as well. Driven by the desires to obtain collegiate scholarships or potentially earn a professional contract, youth basketball players are increasingly becoming highly competitive [25]. It is therefore not surprising that the prevalence of PTP has increased accordingly. 

Some studies have investigated the prevalence of ATP in youth sports. Cassel et al. [26], in a cross-sectional study, reported an overall ATP prevalence of 1.8% (2.0% in males and 1.6% in females) in 760 adolescent athletes from 16 different sports (basketball not included). In another study, a prevalence of 7.5% was reported for high school runners aged 13–18 years [27]. Further, Emerson et al. reported a prevalence of 12.5% and 17.5% in male and female gymnasts, respectively [28]. Although the prevalence estimate reported for ATP in our study (i.e., 4.5% overall) is higher than the average reported for 16 popular youth sports examined by Cassel et al. [26], it is much lower than the prevalence reported for adolescent runners and gymnasts [28]. It is thus conceivable that ATP is less common in youth basketball, especially when compared with its prevalence in other youth sports involving repeated jumping and landing. 

### 4.2. Time to Tendinopathy, Symptoms’ Duration and Severity Score

The importance of extensively appraising injury burden to fully understand injury risk has been emphasized [9]; it provides a complete picture of risk and gives insight of injury morbidity and potential consequences. Further to prevalence estimates, we examined other measures of burden to elucidate the problem of PTP and ATP in youth basketball players. Results from our study suggest that 50% of asymptomatic players at the start of the season may develop PTP at approximately 7 weeks (8 weeks for males and 6 weeks for females) into the season. This finding has direct implications for practice. For example, effort to reduce the burden of PTP may include decisions by sport directors/club administrators and basketball coaches regarding the need and timing for a periodic in-season PTP evaluation and subsequent workload adjustments in competitive youth basketball players based on the sex-specific estimated time to tendinopathy reported in the current study. 

Although the prevalence of ATP was found to be low, it appears the burden of ATP on players as measured by both symptoms’ duration proportion and severity scores was high and comparable to that of PTP. This finding corroborates previous report indicating a high burden of ATP in elite basketball players [2]. An overall median symptoms duration proportion of 83% for PTP and 75% for ATP found in the current study suggests that youth basketball players have chronic tendinopathies [3] with symptoms lasting through an extensive period of a competitive season. Future research should investigate the short-term impact of injury chronicity on players’ overall health, e.g., psychological wellbeing and players’ quality of life.

### 4.3. Risk Factors of Patellar and Achilles Tendinopathy

Based on a cluster-adjusted multivariable logistic regression analysis, our study suggested that sex and previous anterior knee pain are significantly associated with PTP risk in youth basketball players. In accordance with previous studies in basketball and volleyball, age was not found to be a significant risk factor for PTP in the current study [1,29]. Of note, the odds of having PTP in players who reported previous anterior knee pain (within past 3 months) at baseline was 8.5-fold compared with players without previous anterior knee pain. Anterior knee pain in basketball players would most likely be a quadricep tendinopathy, patellar tendinopathy, or patellofemoral pain syndrome. A pre-participation screening program that includes assessment for recent history of anterior knee pain (specifically including the aforementioned) prior to a competitive season may be valuable in identifying youth basketball players at high risk of PTP. 

Our finding of male sex as a risk factor for PTP corroborates that of Cook et al. [8] and other studies in adult elite and recreational basketball players [1,29]. Although the mechanism by which male players are more susceptible to PTP than female players is not fully known, it is suggested that estrogen may have a protective function in females [30]. 

Multivariable regression analyses could not be run for ATP given the low prevalence of ATP. Similarly, the total number of participants and events in the sub-cohort with performance data were inadequate for risk factor analysis in a multivariable logistic regression. 

### 4.4. Methodological Considerations

As it was impracticable for our study physical therapist to evaluate all 515 players for PTP and ATP on a weekly basis, we implemented a self-report measure of PTP using a questionnaire whose diagnostic accuracy has been validated against clinical examination for PTP [13]. Previous studies reporting the prevalence of PTP or ATP based on a questionnaire or pain mapping approach have used tools that were not pre-validated [27,29]. As used in the current study, a self-report methodology with calculations of prevalence, rather than incidence, has been shown to be more accurate in reporting overuse injuries [11,12]. In contrast to previous studies that used the OSTRC Overuse Injury Questionnaire to report average weekly/bi-weekly prevalence and severity scores [24,31,32,33], we chose to examine the trend of weekly PTP and ATP through line graphs. This is because the repeated measures of weekly prevalence are not independent, and as such, mean (95% CIs) or median (IQR) are likely to yield erroneous summary estimates. Furthermore, we used a robust cluster-adjusted multivariable regression analysis to evaluate independent risk factors for PTP risk, a strength in this study. 

### 4.5. Study Limitations 

This study had some limitations. First is the possibility of response and recall bias that might have impacted our prevalence estimates of PTP and ATP. Although the overall response rate for our primary outcomes, season prevalence of PTP, and season prevalence of ATP was adequate, the information collected at follow-up through phone interviews might have been impacted by recall bias. 

Another limitation is that we are unable to ascertain the validity of the Achilles tendinopathy part of the OSTRC-Tendinopathy Questionnaire, as this is yet to be evaluated in the literature. Given that the questions in the ATP part of the questionnaire are similar to the ones in the PTP part, we speculate that the diagnostic accuracy of ATP segment might be similar to that of the patellar tendinopathy part of the OSPTC Tendinopathy Questionnaire. 

Third, although we used the term risk factors to define the independent variables examined in the risk factor analysis conducted in our study, we are unable to confirm a causal relationship between PTP and these variables. For any causal relationship to be established, it is critical that exposure precedes event or disease among other conditions [34]. In our study, 22% of players indicated symptoms of anterior knee pain at baseline. It is speculated that many of these 22% had PTP at baseline. We decided a priori not to exclude such players, given that our measure of risk was prevalence (sensitive measure for chronic/overuse injuries) and not incidence (sensitive for acute onset injuries) [11]. While we acknowledge the importance of temporality, we believe that its application in risk evaluation for chronic injuries is restricted. Additionally, excluding symptomatic players at baseline in a bid to establish temporality in our study would result in a biased sample frame, which would in turn limit the external validity of our finding [11,18].

Finally, we did not evaluate players’ workload in the present study. Workload may be a key risk or protective factor for PTP and ATP [35,36]. The complex system approach for pattern recognition and risk profiling of sports injury etiology may serve as an innovative next-step towards the prevention and control of PTP and ATP [37,38,39]. We recommend that future studies evaluate the relationship between workload and PTP while considering the potential moderating effects of other risk factors for PTP. 

## 5. Conclusions

The prevalence and burden of patellar tendinopathy is high in competitive male (11–18 years) and female (13–18 years) youth basketball players; 1 in 4 male players and 1 in 5 female players reported symptoms of patellar tendinopathy in a competitive season. Although less common, the burden of Achilles tendinopathy appears significant and comparable to that of patellar tendinopathy. Risk factors of patellar tendinopathy in youth basketball include sex and previous anterior knee pain. The current findings have implications for practice and future research relating to the prevention and in-season management of patellar and Achilles tendinopathy in youth basketball. There is a crucial need for countermeasures to abate the risk and potential consequences of both patellar and Achilles tendinopathy in competitive youth basketball. This includes raising awareness about the burden of patellar and Achilles tendinopathy among stakeholders—administrators, coaches, players, and parents—and promoting the uptake of current best practices for both primary and secondary prevention of patellar and Achilles tendinopathy, including progressive tendon loading that incorporates both isometric and isotonic exercises [40,41,42,43].

## Figures and Tables

**Figure 1 ijerph-18-09480-f001:**
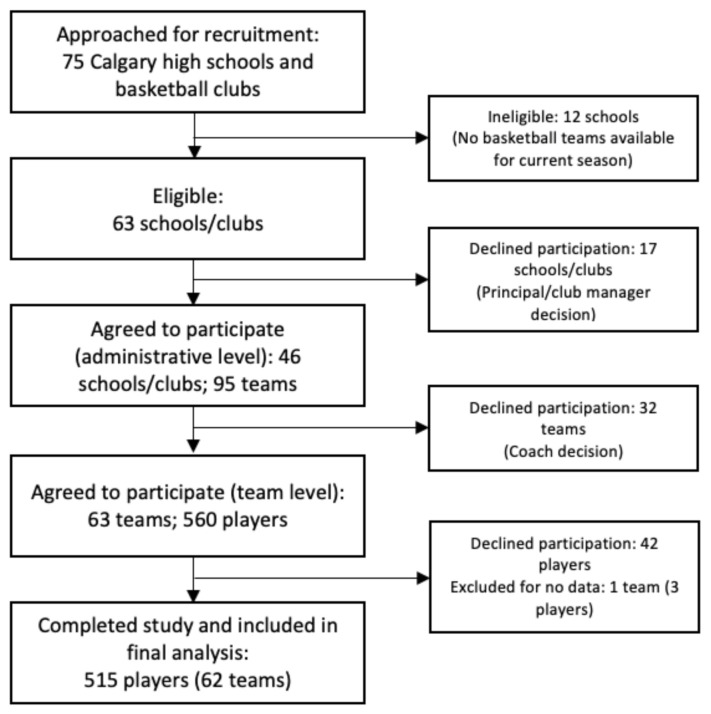
Flow chart of players through the study.

**Figure 2 ijerph-18-09480-f002:**
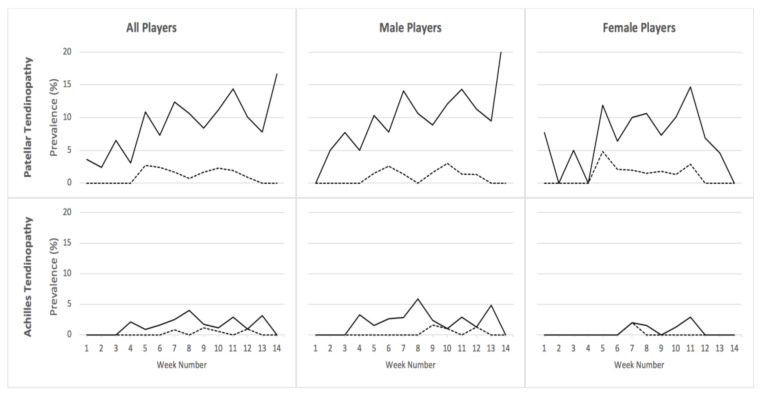
Prevalence of all patellar and Achilles tendinopathy (straight line, **^_____^**) and substantial patellar and Achilles tendinopathy (broken line, -----) in all players, male and female players, over a 14-week follow-up period. Participant numbers varied from 12 players in week 14 to 178 players in week 9. Week numbers were sequentially aligned for both school and club seasons; a 1-week Christmas break (week 4) was absent for the school season, i.e., week 4 prevalence is based on the club season only. Substantial tendinopathy connotes tendinopathy leading to moderate or severe reductions in training volume or performance, or an inability to participate.

**Table 1 ijerph-18-09480-t001:** Player baseline characteristics by sex.

	Males *n* = 315	Females *n* = 200
**Age (years)**		
Median (range)	16 (11–18)	16 (13–18)
Missing, *n* (%)	59 (19)	31 (16)
**Height (cm)**		
Median (range)	179 (110–201)	168 (152–193)
Missing, *n* (%)	32 (10)	14 (7)
**Weight (kg)**		
Median (range)	67 (38–132)	60 (41–141)
Missing, *n* (%)	30 (10)	15 (8)
**Previous Knee Injury (1 Year), *n* (%)**		
Yes	216 (69.6)	152 (76.0)
No	30 (9.5)	30 (15.0)
Missing	69 (21.9)	18 (9.0)
**Previous Anterior Knee Pain (3 months) *n* (%)**		
Yes	90 (28.6)	53 (26.5)
No	141 (44.7)	116 (58.0)
Missing	84 (26.7)	31 (15.5)
**Previous Achilles Tendon Pain (3 months) *n* (%)**		
Yes	191 (60.6)	150 (75.0)
No	38 (12.1)	14 (7.0)
Missing	86 (27.3)	36 (18.0)
**Player Position, *n* (%)**		
Guard	136 (43.2)	89 (44.5)
Post	50 (15.9)	45 (22.5)
Combo	81 (25.7)	52 (26.0)
Missing	48 (15.2)	14 (7.0)
**Single Sport Baskeball Participation, *n* (%)**		
Yes	101 (32.1)	70 (35.0)
No	161 (51.1)	99 (49.5)
Missing	53 (16.8)	31 (15.5)

**Table 2 ijerph-18-09480-t002:** Multivariable Logistic Regression Model for Patellar Tendinopathy in youth basketball players (*n* = 515).

Variable	OR	95% CI	*p*-Value
Sex			
Female (Referent)	-	-	-
Male	2.23	1.10–4.69	0.026 *
Previous Anterior Knee Pain			
No (Referent)	-	-	-
Yes	8.79	4.58–16.89	<0.001 *
League Setting			
Club (Referent)	-	-	-
School	0.76	0.39–1.51	0.436
Weight	0.98	0.96–1.01	0.156
Age	1.27	0.96–1.67	0.090

Model adjusted for clustering by team and other covariates (the following covariates were included at some point: height, previous knee injury, playing position, and basketball specialization). The final model included the covariates presented in this table, representing the most parsimonious model (based on the need to adjust for the 13 participants in both league seasons and the clinical significance of weight and age), regardless of statistical significance. CI, confidence interval; OR, odds ratio. * Statistical significance at alpha = 0.05.

## Data Availability

Data is available upon reasonable request.

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
