# Peer review of "The Burden and Risk Factors of Patellar and Achilles Tendinopathy in Youth Basketball: A Cohort Study"

_ijerph, 2021, doi:10.3390/ijerph18189480_

Round 1
Reviewer 1 Report
Thank you for the opportunity to review your manuscript. I enjoyed reading your work. My minor suggestions/comments are below for your consideration.
What were the total number of weeks for the high school and club basketball season? could this be made clearer in the methods
were these separate cohorts or did some of the youth play in both 'seasons'?
how did you determine the cut-off for the less than expected responses on the OSTRC questionnaire?
did you record previous AT injury (1y and 3 months) at baseline?
could you add a reference for elite adult male prevalence line 279?
are there specific differences between your study and cook et al that you can elaborate on with regard to why your results are so different?
is there a reference for the changes in/increase in frequency and intensity of participation mentioned?
time to onset of symptoms - any literature regarding the onset timeline (7weeks?) do the included athletes complete any off-season training prior to the playing season? If yes/no what are your thoughts on the role of prehabilitation? is it a case of underload and subsequent overload??
you suggested workload adjustment, do you have any recommendations regarding future studies looking at workload?
beyond screening for prior pain and increased risk of in-season tendonopathy - what could this info inform/change for those identified as an increased risk? what should coaches/athletes/parents ideally do with this info? what is the current best practice could you reference a key piece of literature for the reader?
good representation of limitations
as per my comment above perhaps lines 381-386 fits better in the time to onset paragraph
Author Response
Thank you for the opportunity to review your manuscript. I enjoyed reading your work. My minor suggestions/comments are below for your consideration.
Response: Thank you for your kind words, and for your time in reviewing our manuscript.
What were the total number of weeks for the high school and club basketball season? could this be made clearer in the methods
Response: The total number of weeks for high school and club basketball season varied among teams as this depended on whether they qualified for playoffs and final rounds. This is the reason we stated the range of time in our methods section. In addition to this, we have added the average number of weeks for the entire cohort in our result for clarification.
Lines 254-255: A total of 98 players reported at least one episode of PTP through the league season (spanning an average of 9.5 weeks)…
were these separate cohorts or did some of the youth play in both 'seasons'?
Response: They were separate cohorts; however, 13 participants played in both seasons. This is the reason we adjusted for league season as a variable and kept it in our multivariable model.
how did you determine the cut-off for the less than expected responses on the OSTRC questionnaire?
Response: This was based on the post-season coach and online information available to us at the time. We have clarified this in our methods.
Lines 134-140: Determined based on teams’ final league schedules as available online for the 2016/2017 league seasons, players with less than the expected number of OSTRC Tendinopathy Questionnaire responses through the competitive season (i.e., 10 – 12 weeks of high school basketball season and 8 – 14 weeks of club season; depending on whether player/team made playoffs) were followed up post-season through a telephone interview by the study physical therapist or trained research assistants to complete the OSTRC Tendinopathy Questionnaire for any missing weeks.
did you record previous AT injury (1y and 3 months) at baseline?
Response: No for 1 year, but yes for 3-month history. We have added the baseline data for 3-month history for previous Achilles tendon pain (please see Table 1).
could you add a reference for elite adult male prevalence line 279?
Response: Done
Lines 320-322: An overall PTP prevalence of 19 % (23% in males and 13% in females) presented in our study is lower than the prevalence (32%) reported for male elite adult basketball players [1].
are there specific differences between your study and cook et al that you can elaborate on with regard to why your results are so different?
Response: We have expounded on this.
Lines 326-330: For example, in our study, we prospectively examined PTP weekly (repeated measurements) through self-report (which is highly sensitive in capturing gradual onset injuries) [22,23] to derive our main outcome of season prevalence of PTP while in the study by Cook et al., PTP prevalence was based on a cross-sectional examination (one-time measurement) by a clinician.
is there a reference for the changes in/increase in frequency and intensity of participation mentioned?
Response: No, this is more of a perception based on our field experience. We have modified the sentence to reflect this.
Lines 330-332: Further to this, the frequency and intensity of participation “appears” to have increased over the past decade, and it is very likely gradual onset/overuse injuries have increased as well.
time to onset of symptoms - any literature regarding the onset timeline (7weeks?) do the included athletes complete any off-season training prior to the playing season? If yes/no what are your thoughts on the role of prehabilitation? is it a case of underload and subsequent overload??
Response: Thanks for your great thoughts. We believe our study is the first to note this finding of average of 7 weeks to PTP onset; as such, we did not have any literature to compare this finding with. Our experience with the youth teams is that they do not have any formal off/preseason training; however, some of them engage in off-season basketball. It is possible that many participants that had PTP did not have on-going or pre-season training/basketball exposure.
you suggested workload adjustment, do you have any recommendations regarding future studies looking at workload?
Response: Yes, we mentioned this in our last paragraph.
Lines 429-434: Finally, we did not evaluate players’ workload in the present study. Workload may be a key risk or protective factor for PTP and ATP [36,37]. The complex system approach for pattern recognition and risk profiling of sports injury etiology may serve as an innovative next-step towards the prevention and control of PTP and ATP [38–40]. We recommend that future studies evaluate the relationship between workload and PTP while considering the potential moderating effects of other risk factors for PTP.
beyond screening for prior pain and increased risk of in-season tendonopathy - what could this info inform/change for those identified as an increased risk? what should coaches/athletes/parents ideally do with this info? what is the current best practice could you reference a key piece of literature for the reader?
Response: Thank you for your comment. We have included additional thoughts in our conclusion.
Lines 447-452: This includes raising awareness about the burden of patellar and Achilles tendinopathy among stakeholders – administrators, coaches, players and parents – and promoting the uptake of current best practices for both primary and secondary prevention of patellar and Achilles tendinopathy, including progressive tendon loading that incorporates both isometric and isotonic exercises [42–45].
good representation of limitations
Response: Thank you.
as per my comment above perhaps lines 381-386 fits better in the time to onset paragraph
Response: Thank you for this suggestion. We decided to keep the paragraph in the limitation section since it is a study limitation.
Reviewer 2 Report
In general, this is a well written paper. I only have several specific comments.
First, it is necessary to briefly introduce the training and competitive mathches that young basket players need to take in this study. This is very important for global readers since many other countries do not have basket clubs and regular basketball seasons for the youths.
Second, as the authors mentioned, this study was potentially biased by the assessment of primary outcome. Self-reports cannot replace the diagnoses made by the doctor, espcially for clinically important symtoms.
Third, please add the reasons for the selection of potential risk factors.
Fourth, it is necessary to mention the risk factors in the findings of the abstract.
Last, the conclusion could be more specific based on the key findings.
Author Response
In general, this is a well written paper. I only have several specific comments.
Response: Thank you for your time in reviewing our manuscript.
First, it is necessary to briefly introduce the training and competitive mathches that young basket players need to take in this study. This is very important for global readers since many other countries do not have basket clubs and regular basketball seasons for the youths.
Response: Thank you for your note, but our opinion is that readers can get a whole lot of information online if they are interested in learning more about the youth basketball league settings in Calgary, Alberta.
Second, as the authors mentioned, this study was potentially biased by the assessment of primary outcome. Self-reports cannot replace the diagnoses made by the doctor, especially for clinically important symptoms.
Response: We appreciate your concern about self-report of clinical symptoms and its potential bias. While the OSTRC-P is not perfect, its validity has been tested against clinical exam and shown to be comparable to clinical evaluation by a physical therapist (citation in manuscript). Considering the design adopted in our study, it is impracticable to have a physical therapist or physician examine 515 players weekly; reason why we used a self-report measure.
Lines 123-125: …the OSTRC-Patellar Tendinopathy Questionnaire, has been validated against clinical evaluation in youth basketball players with overall sensitivity of 79% and specificity of 98% [13].
Third, please add the reasons for the selection of potential risk factors.
Response: We have added this information.
Lines 181-182: Our choice of potential risk factors was informed by previous literature [10,20].
Fourth, it is necessary to mention the risk factors in the findings of the abstract.
Response: Done.
Lines 41-42: Risk factors include sex and previous anterior knee pain.
Last, the conclusion could be more specific based on the key findings.
Response: Thank you for your comment. We have expanded our conclusion to include specific applications for practice.
Lines 447-452: This includes raising awareness about the burden of patellar and Achilles tendinopathy among stakeholders – administrators, coaches, players and parents – and promoting the uptake of current best practices for both primary and secondary prevention of patellar and Achilles tendinopathy, including progressive tendon loading that incorporates both isometric and isotonic exercises [42–45].